# Development of the Composite Thermoluminescent Detectors Based on the Single Crystalline Films and Crystals of Perovskite Compounds

**DOI:** 10.3390/ma15238481

**Published:** 2022-11-28

**Authors:** Sandra Witkiewicz-Lukaszek, Anna Mrozik, Vitaliy Gorbenko, Tetiana Zorenko, Pawel Bilski, Yurii Syrotych, Yuriy Zorenko

**Affiliations:** 1Institute of Physics, Kazimierz Wielki University in Bydgoszcz, Powstańców Wielkopolskich Str., 2, 85090 Bydgoszcz, Poland; 2Henryk Niewodniczański Institute of Nuclear Physics, Polish Academy of Sciences, 152 Radzikowskiego Str., 31342 Cracow, Poland

**Keywords:** perovskites, single crystals, single crystalline films, liquid phase epitaxy, thermoluminescence

## Abstract

This work is dedicated to the development of new types of composite thermoluminescent detectors based on the single crystalline films of Ce-doped GdAlO_3_ perovskite and Mn-doped YAlO_3_ and (Lu_0.8_Y_0.2_)AlO_3_:Mn perovskites as well as Ce and Pr-doped YAlO_3_ single crystal substrates. These detectors were obtained using the Liquid Phase Epitaxy growth method from the melt solution based on the PbO-B_2_O_3_ fluxes. Such composite detectors can by applied for the simultaneous registration of different components of mixed ionization fluxes using the differences between the thermoluminescent glow curves, recorded from the film and crystal parts of epitaxial structures. For creation of the new composite detectors, we considered using, for the film and crystal components of epitaxial structures (i) the different perovskite matrixes doped with the same type of activator or (ii) the same perovskite host with various types of activators. The thermoluminescent properties of the different types of epitaxial structures based on the abovementioned films and crystal substrates were examined in the conditions of β-particles and X-ray excitation with aim of determination of the optimal combination of perovskites for composite detectors. It was shown that, among the structures with all the studied compositions, the best properties for the simultaneous thermoluminescent detection of α- and X-rays were the GdAlO_3_:Ce film/YAlO_3_:Ce crystal epitaxial structure.

## 1. Introduction

Unlike scintillation detectors, which record ionization radiation and display results continuously (active mode), passive thermoluminescent TL radiation detectors accumulate signal over the entire course of their exposure. Namely, TL detectors record the total absorbed dose of ionizing radiation by the way of localization of the respective quantity of charge curriers at trapping centers, connected usually with defect centers of TL material [1,2,3]. During the registration stage, the TL reader heats the composite samples and liberates electrons/holes from their traps, which results in their recombination at emission centers created by special kinds of impurities or defects [4]. This process releases thermostimulated emission (TSL) light in the visible or UV ranges, which is measured by the TL reader. The amount of TSL light is proportional to the total absorbed radiation dose accumulated by the TL over the duration of its exposure [1,2,3,4].

The most common TL detectors are those based on lithium fluoride [4,5], but new materials and new applications are still under development [6]. In the past, some types of thin-film TL detectors for the measurement of weakly penetrating radiation were developed as well [7,8,9,10].

In addition to the so-called composite scintillators of the “phoswich-type” (phosphor sandwich)*,* for the simultaneous registration of the components of mixed ionization fluxes (for example α-and β- particles and X rays or γ-quanta), the composite TL detectors can be also successfully applied [11,12,13]. The development of such detectors of both types is now an actual subject of luminescent material engineering [12,13,14,15]. The basis for such engineering of composite detectors is the latest subject in the development of bulk single crystal (SC) and single crystalline film (SCF) scintillators and TL materials as well as the technologies of their production using the Czochralski and other melt growth methods and the liquid phase epitaxy (LPE) growth techniques, respectively [11,12,13,14,15].

In our previous works [12,13], we considered, for the first time, the creation of composite TL materials based on the LPE grown SCF/SC epitaxial structures of several garnet compounds for the passive registration of the incoming ionization fluxes. In these works, the simultaneous registration of different components of mixed ionization fluxes occurs using differences in the TL glow curves coming from the SCF and SC parts of the composite TL detector under α-and β-particle excitation (Figure 1). To characterize the rate of α/β particle scintillation decay time (∆T) separation, we used the difference ∆T = T_SCF_ − T_SC_ between the position of the main TL peaks, corresponding to the SCF (T_SCF_) and SC substrate (T_SC_) parts of the composite detector. Namely, for the composite TL detectors, based on the LPE grown epitaxial structures of Ce^3+^-doped Lu_3_Al_5_O_12_ (LuAG) and Y_3_Al_5_O_12_ (YAG) garnets, the significant differences in the positions (up to 80 °C) and intensity (more than by one order of magnitude) of the main TL peaks were observed after α- and β-particle irradiation [12]. Furthermore, the difference ∆T between the main peaks of TL glow curves can increase even up to up to 150–215 degrees for Lu_3−x_Gd_x_Al_5_O_12_:Ce SCFs/YAG:Ce SC epitaxial structures at changing Gd concentrations in the SCF part in the x = 2–1.5 range [13].

This work presented the next attempt in creation of composite TL detectors in the form of the LPE-grown epitaxial structures based on SC and SCF oxide compounds. Specifically, we considered the two approaches for the creation of a composite TL detector: (a) using different matrices of TL materials doped with the same type of activator; and (b) using various types of activators and the same matrix for the SC and SCF parts of the detector (Figure 1).

For the realization of these approaches, we considered the production of the epitaxial structures of the perovskite materials based on the rare-earth and transition metal-doped SCF and SC of RAlO_3_ (R = Gd, Y) perovskite compounds [16,17]. In the frame of these approaches, the epitaxial structures based on the SCF of Mn-doped YAlO_3_ perovskite and Ce or Pr-doped YAP SC substrates as well as the epitaxial structures based on the Ce-doped GdAlO_3_ SCF and Ce or Pr-doped YAP SC substrates, respectively, were created using the LPE growth method from the melt solution based on the PbO-B_2_O_3_ fluxes.

In spite of the large effective atomic number in comparison with the respective garnet compounds, based on the R-Al cations [11], the perovskite materials can be better candidates to create composite TL detectors and composite scintillators as well [18,19,20,21]. It is also important to note here that the YAP:Ce crystals, and other doped SC crystals of the YAP family, show suitable TL properties after different kinds of ionization radiation [18,19]. Namely, the SC and powders of YAP:Mn have been considered for many years as effective media for TL detectors [22,23,24,25,26]. Furthermore, SCF of YAP:Mn perovskite possesses the suitable TL properties for the detection of X quanta [27]. In this work, we tried to connect the properties of several YAP-based crystals and films, suitable for the creation of the composite TL detectors (Figure 1).

## 2. Materials and Methods

Composite TL detectors based on the GdAP:Ce SCF/YAP:Ce SC, GdAP:Ce SCF/YAP:Pr SC and YAP:Mn SCF/YAP:Ce SC, YAP:Mn SCF/YAP:Pr SC, (Lu_0.2_Y_0.8_):Mn SCF/YAP:Ce SC epitaxial structures were crystalized using the LPE method from the melt-solution based on PbO-B_2_O_3_ flux (Figure 2). The growth conditions are presented in Table 1.

The concentration of dopants in substrate and SCF samples was measured using a SEM JEOL JSM-820 electron microscope equipped with a microanalyzer EDS with IXRF 500 and LN2 Eumex detectors. The Ce content in YAP:Ce and YAP:Pr crystal substrates was 0.25 and 0.5 at. %, respectively. The Ce and Mn contents in SCF samples show the proportional dependence on 1/T_g_ value, where T_g_ is the temperature of film growth. Namely, the Ce^3+^ concentration in GdAP:Ce SCFs was changed in the 0.065–0.1 at. % range depending on growth temperature T_g_ in the 975–985 °C range (Table 1). The Mn content in YAP:Mn and Y_0.8_Lu_0.2_AP:Mn SCFs was varying in the 0.07–0.17 at. % range, depending on growth temperature T_g_ in the 975–985 °C range (Table 1).

The surface morphology of the GdAP:Ce SCF/YAP:Ce SC and YAP:Mn SCF/YAP:Ce was measured used a Keyence optical microscope (Figure 3). The surface roughness was measured using two factors: R_a_ measures the average length between the peaks and valleys and the deviation from the mean line on the entire surface within the sampling length. R_z_ measures the vertical distance from the highest peak to the lowest valley within five sampling lengths and averages the distances. As can be seen from Figure 3, a much better surface quality was observed for quasi-epitaxial growth of GdAP:Ce SCF/YAP:Ce SC epitaxial structure in comparison with the homoepitaxial growth of the YAP:Mn SCF /YAP:Ce SC counterpart. Namely, for the high-quality GdAP:Ce SCF/YAP:Ce SC sample, the values were R_a_ = 0.24 μm, R_z_ = 1.23 μm for the low-quality YAP:Mn SCF/YAP:Ce SC: R_a_ = 1.21 μm, R_z_ = 6.84 μm. This untypical result is likely connected with the difficulties in the homogeneous growth of Mn-doped YAP SCF from melt-solutions with high concentration on MnO_2_ activator caused by low solubility of Mn ions in perovskite host due to their very low segregation coefficient [27].

For characterization of the luminescent properties of the SCF and SC parts of the composite TL detectors, the cathodoluminescence (CL) spectra were used. The CL spectra were measured using a SEM JEOL JSM-820 electron microscope additionally equipped with a Stellar Net spectrometer working in the 200–1100 nm range.

The emission spectra of the studied materials extend to the long wavelengths, reaching 800 nm. For this reason, it was not possible to use for TL measurements the standard TL readers (such as Harshaw 3500 or Risø DA-20), whose spectral sensitivities are optimized for UV/blue emission and which are not sensitive beyond c.a. 500 nm. Instead, the thermoluminescence measurements were carried out in the two-dimensional TL reader constructed at IFJ PAN [28]. The reader is equipped with a PCO SensiCamTM VGA CCD 12-bit camera with spectral range 300 nm to 800 nm. The temperature was raised linearly up to the temperature of 400 °C. The measured TL glow curves were deconvoluted into individual peaks using the GlowFit code [29].

Investigations of properties of composite detectors required the application of both weakly and strongly penetrating radiation. The first one included soft X-rays. The mini X-ray generator (needle-like anode X-ray tube) with a Be window was produced in the National Centre for Nuclear Research in Warsaw [30]. The electric voltage of the mini X-ray generator was set to 15 kV. The average energy of X-rays was assumed to be 10 keV.

Some of the values of photon absorption for the investigated materials are shown in Table 2. The value of photon absorption for each sample was calculated using Formula 1, where *μ/ρ* is the mass attenuation coefficients and *d* is the mass thickness. All of the *μ*/*ρ* values were taken from the NIST database (https://physics.nist.gov/PhysRefData/XrayMassCoef/tab3.html), accessed on 21 November 2022.
(1)$$\frac{I}{I_{0}}\;=\;\exp(-\left(\frac{\mu }{\rho }\right)\times d),$$

Irradiations with strongly penetrating radiation were performed using ^90^Sr/^90^Y beta source with 1.4 GBq activity.

## 3. Optical Properties

### 3.1. CL Spectra of SCF and Substrates

The CL spectra of GdAP:Ce SCF/YAP:Ce SC structures (Figure 4a) show that the dominant doublet luminescence bands in the UV range peaked at 360–380 nm, connected with the 5d (^3^T_2g_)-4f (^2^F_5/2,7/2_) transitions of Ce^3+^ ions. The broad emission band in the visible range observed in GdAP:Ce SCF/YAP:Ce SC is caused by the luminescence of centers related to the Pb^2+^ lux impurity [16]. Namely, such a band is probably caused by the emission of excitons localized around Pb^2+^ ions or more complex Pb^2+^ based centers in the SCFs of perovskites [16]. Other sharp emission bands of GdAP:Ce SCF/YAP:Ce SC peaked at 545 nm and at 592 nm and 615 nm in the visible range are related to the luminescence of Tb^3+^ and Eu^3+^ trace impurities, respectively [19].

The CL spectra of YAP:Mn SCF samples (Figure 4b) are a superposition of the luminescence of the different valence states of Mn ions. The dominant luminescence of Mn^3+^ ions in the broadband peaked approximately at 660 nm corresponding to the ^5^T_2_ → 5^E^ transitions. In the CL spectra we also noted the weak luminescence of Mn^2+^ ions in the cub-octahedral position of Y^3+^ cations in the band peak at 552 nm (^4^T_1_ → ^6^A_1_ transitions).

The CL spectra of YAP:Ce SC substrate (Figure 4c) present the dominant luminescence band of Ce^3+^ ions peaked at 366 nm, caused by the 5d → 4f (^2^F_5/2,7/2_) radiation transitions. The low intensity broadband peaked approximately at 600 nm range is caused by the luminescence of dimer or more complex centers based on the charged oxygen vacancies. The sharp lines in the visible range are caused by Tb^3+^ trace impurity.

The CL spectra of the YAP:Pr, Ce SC substrate showed that the UV emission bands peaked at 245 nm and 275 nm corresponding to the 5d →4f (^3^H_4_, ^3^H_5_, ^3^H_6_, ^3^F_3(4)_) transitions of Pr^3+^ ions and sets of sharp lines in the visible range with the main peaks at 492 nm and 621 nm connected with 4f–4f radiation transitions from ^3^P_0_ and ^1^D^2^ levels. The low-intensity band that peaked at 374 nm is probably caused by the Ce^3+^ trace impurity.

### 3.2. Thermoluminescence

The thermoluminescence of GdAP:Ce SCF/YAP:Ce SC,YAP:Mn SCF/YAP:Ce SC, YAP:Mn SCF/YAP:Pr,Ce SC, and LuYAP:Mn SCF/YAP:Ce SC epitaxial structures was measured after irradiation by X-rays and beta-rays from ^90^Sr/^90^Y sources. Before each irradiation procedure the samples were preheated up to 400 °C to erase the possible TL information due to previous irradiation.

It is important to mention that beta particles passing through SCF also generate a signal in the surface layer. Thus, the TL glow curve after irradiation with ^90^Sr/^90^Y source is a sum of the signals from SCF and substrate. The signal contribution from the SCF in this case is probably minor. Similarly, in the case of irradiating the samples with layers thicknesses of 10.5 and 17 µm using the soft X-rays, only 20%–30% of the X-ray photons will be absorbed in the film (see Table 2) and the remaining part moves to the substrate. So, the TL curves after irradiation with X-rays and beta particles are similar, but under more detailed analysis, it is possible to see the differences.

#### 3.2.1. Yttrium Perovskites

The results of deconvolution analysis of YAP:Mn/YAP:Ce (#1, #2), LuYAP:Mn/YAP:Ce (#3) and YAP:Mn/YAP:Pr (#4) epitaxial structures after exposure to X-rays and β-particles are presented in Figure 5 and Table 3. The luminescence of Mn^3+^ ions corresponds to the main peak at 130–150 °C and the luminescence of Mn^2+^ ions corresponds to 189–197 °C peak [27].

In the case of samples #1 and #2 there were no differences in the glow-curve shape between irradiations with X-rays and beta-particles. This may suggest that most of the signal originates from the substrate. In the case of the LuYAP:Mn/YAP:Ce (sample #3), there was a significant difference concerning a relative increase of the high-temperature peak at 235–238 °C for X-ray irradiation. This indicates that this peak is mainly related to the SCF. The most significant differences can be observed for the sample based on Pr-doped substrate (#4). For X-ray exposure, all high-temperature peaks were much enhanced and the peak at 303 °C was dominant of the whole glow-curve.

#### 3.2.2. Gadolinium Perovskites

In Figure 6 the set of results of the deconvolution of TL glow curves for GdAP:Ce/YAP:Ce (#5 and #6) and GdAP:Ce/YAP:Pr (#7) after irradiations with X-ray and beta-particles are presented. The most noticeable differences between the two TL curves (after irradiation with two different radiation types) can be seen in the 200–270 °C range. The TL peaks were shifted to higher temperature by c.a. 8–17 °C after X-ray irradiation. As presented in Table 2, most of the 10 keV photons (X-rays) were absorbed in SCF (depends on the SCF thickness). It seems therefore probable that for all samples irradiated with X-rays, the peaks in c.a. 260 °C originate from SCF. It can also be recognized that for all of the analyzed samples, the peak in the area 200–270 °C was wider after irradiation with beta particles than after irradiation with X-rays. This can suggest that after irradiation with beta particles, this peak is, in fact, a sum of two peaks which originate from SCF and SC. This may indicate differences between the SCF and substrate signals. Table 4 presents set of parameters for single peaks of all of the analyzed TL glow curves.

## 4. Discussion

In Figure 7 the TL glow curves of the GdAP:Ce SCF/ YAP:Ce SC epitaxial structure with different SCF thicknesses after irradiation with X-rays (the average energy—10 keV) are shown. The TL glow curves can be divided into four areas (marked in Figure 7). Area III (for X-ray excitation, Figure 7) is the signal from the film and substrate. For the GdAP:Ce SCF/YAP:Ce SC sample with a film thickness of 35 µm where the X-rays are absorbed in 98%, we only observed a signal from the layer. In the thinner samples, the X-rays partially passed to the substrate.

When analyzing the graph in Figure 7, attention should be paid to the peak intensities in regions II and III for GdAP:Ce SCF/YAP:Ce SC epitaxial structures with different SCF thicknesses ranging between 12 and 35 μm. To characterize the differences, the ratio of the TL signal from region II to the signal from region III was used (Figure 8b). From the graph we can see that the ratio decreased when increasing the layer thickness. The decrease can be explained by the fact that x-rays are absorbed in the SCF of the thermoluminescent detector. The absorption of x-rays in GdAP:Ce SCF/ YAP:Ce SC samples compared to their thickness can be seen in the Figure 8a, corresponding to the abovementioned conclusions.

Figure 9 compares the glow-curves after irradiation with ^90^Sr/^90^Y β-particles of the two different types of composite detectors, consisting of YAP:Mn SCF grown on differently doped substrates: YAP:Ce (#2) and YAP:Pr,Ce (#4). For such penetrating radiation, most of the signal is generated in the substrate. The glow-curves varied significantly in the high-temperature region: Pr-doping of the substrates corresponds to the formation of a new peak at 250 °C. It seems that the role of both Ce-doped and Pr-doped substrates were the same. This means that Ce^3+^ and Pr^3+^ ions act as hole traps and the TL glow curves originate from the electron centers only. The difference between both types of ions is seen in the spectral ranges of TL emission of Ce^3+^ and Pr^3+^.

## 5. Conclusions

The possibility of creation of the composite thermoluminescence (TL) detectors based on single crystalline films and single crystals of perovskite compounds, doped with different kinds of dopants, was demonstrated in this work using the LPE growth method. For this purpose, the sets of GdAO_3_:Ce, YAlO_3_:Mn, and Y_0.8_Lu_0.2_AlO_3_:Mn films were LPE grown onto YAlO_3_:Ce and YAlO_3_:Pr crystal substrates.

The development of these types of composite TL detectors could enable a separate registration of the weakly and deeply penetrating radiation components of mixed radiation fluxes, namely X-rays and β-particles. For demonstration of such a possibility, the TL properties of GdAO_3_:Ce film/YAlO_3_:Ce crystal and GdAO_3_:Ce film/YAlO_3_:Pr crystal epitaxial structures as well as YAlO_3_:Mn film/YAlO_3_:Ce crystal, Y_0.8_Lu_0.2_AlO_3_:Mn film/YAlO_3_:Ce crystal, and YAlO_3_:Mn film/YAlO_3_:Pr crystals epitaxial structures were examined under excitation with β-particles and X-rays. For all the mentioned epitaxial structures under study, some differences in the TL glow curves after X-ray and beta-particle irradiations were observed. These differences are mainly concerned the high-temperature range of the TL glow curves.

The most promising combination among the studied films/substrate compositions was the GdAO_3_:Ce film/YAlO_3_:Ce crystal epitaxial structure. For this composition, the intensity of TL peak at 250 °C grew strongly after X-ray irradiation and absorbed mainly in the GdAO_3_:Ce film, in contrast to beta-particle irradiation, absorbed mainly in the YAlO_3_:Ce crystal substrate. This offers a good opportunity for the separation of the TL signals related to the different components of the mixing ionization radiation using this type of composite detector.

## Figures and Tables

**Figure 1 materials-15-08481-f001:**
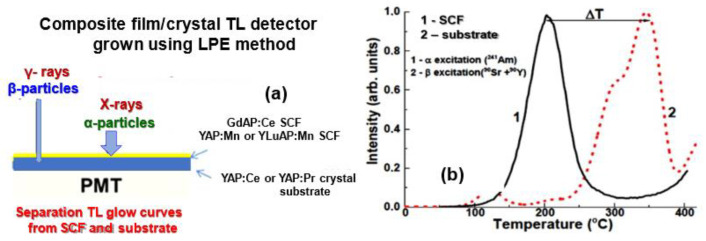
(**a**) Scheme of a two-layer composite TL detector, (**b**) Example of recording ΔT temperature difference between the main peaks in TL glow curves of the film and substrate in a composite TL material.

**Figure 2 materials-15-08481-f002:**
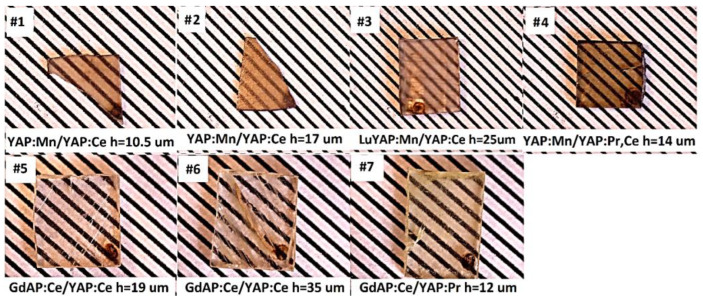
Photos of YAP:Mn SCF/YAP:Ce SC, LuYAP:Mn SCF/YAP:Ce SC, YAP:Mn SCF/ YAP:Pr SC (on the top) and GdAP:Ce SCF/YAP:Ce SC, GdAP:Ce SCF/YAP:Pr SC (on the bottom) composite TL detector.

**Figure 3 materials-15-08481-f003:**
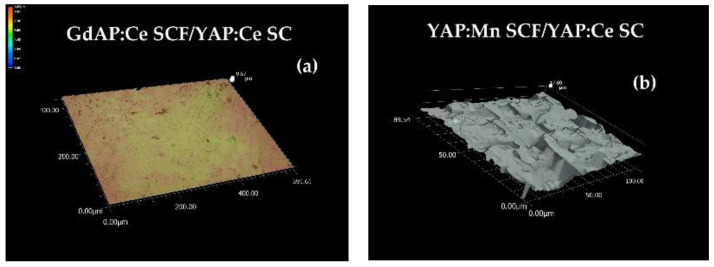
Surface morphology of GdAP:Ce SCF/YAP:Ce SC (**a**) and YAP:Mn SCF/YAP:Ce (**b**) epitaxial structures.

**Figure 4 materials-15-08481-f004:**
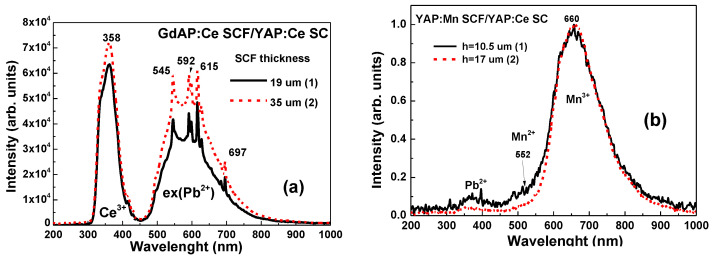
CL spectra of GdAP:Ce SCF/YAP:Ce SC (**a**) and YAP:Mn SCF/YAP:Ce SC (**b**) samples with different SCF thicknesses; (**c**) CL spectra of YAP:Ce SC (1) and YAP,Pr,Ce SC (2) substrates.

**Figure 5 materials-15-08481-f005:**
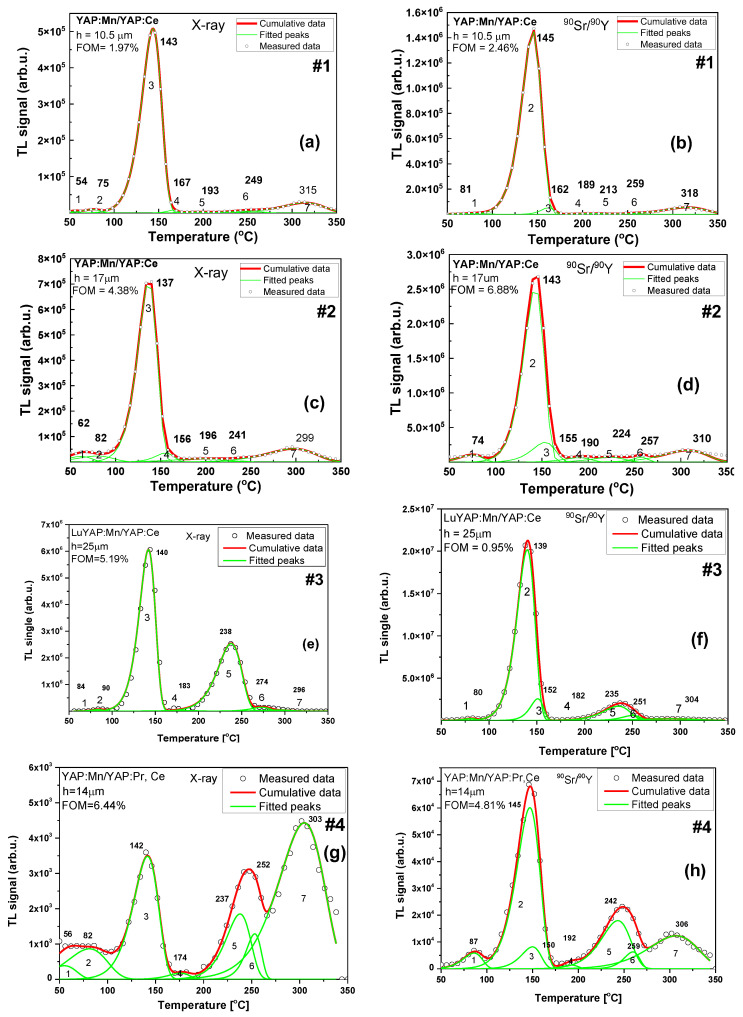
TL glow curves of samples #1, #2, #3, and #4 (**a**–**h**) after excitation by X-rays and β source.

**Figure 6 materials-15-08481-f006:**
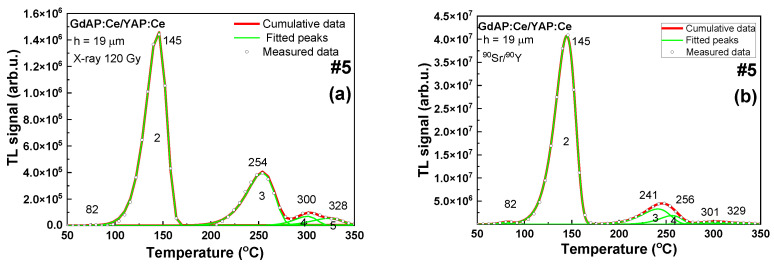
TL glow curves of samples #5, #6, and #7 (**a**–**f**) after excitation by X-rays and β-source.

**Figure 7 materials-15-08481-f007:**
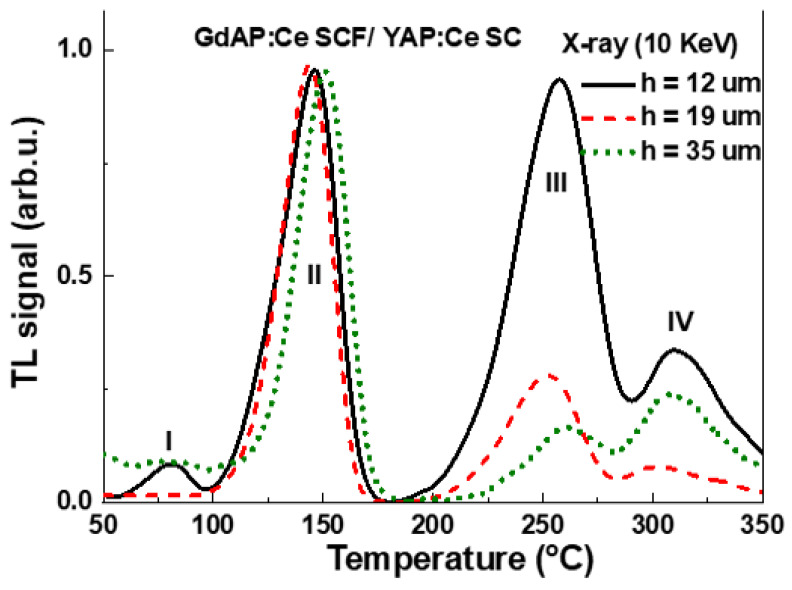
TL glow curves of GdAP:Ce SCF/YAP:Ce SC epitaxial structure under excitation by X-rays (the average energy being 10 keV).

**Figure 8 materials-15-08481-f008:**
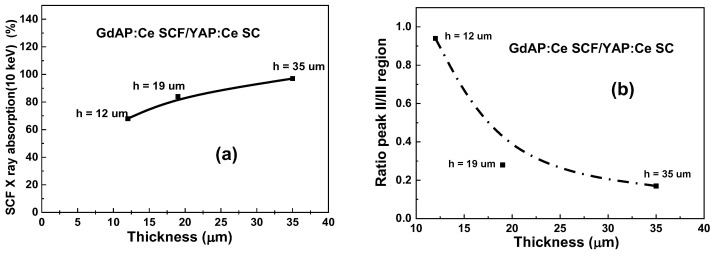
Dependence of X-ray absorption (at energy of 10 keV) on GdAP SCF thickness (**a**) and the ratio between peak TL glow curves from regions II and III in the GdAP:Ce SCF/YAP:Ce SC epitaxial structure (**b**) under excitation by X-rays.

**Figure 9 materials-15-08481-f009:**
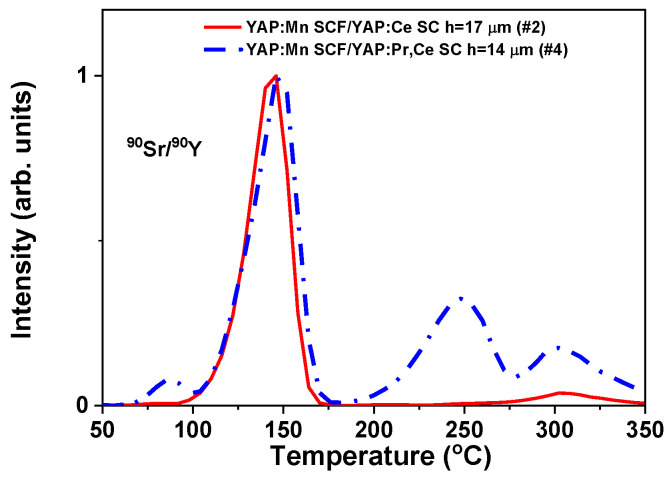
TL glow curves of YAP:Mn SCF/YAP:Ce SC and YAP:Mn SCF/YAP:Pr,Ce SC epitaxial structures under excitation with ^90^Sr/^90^Y β-particles.

**Table 1 materials-15-08481-t001:** Conditions of the LPE growth of GdAP:Ce SCF/YAP:Ce (Pr) SC and YAP:Mn and (YLu)AP:Mn SCFs/ YAP:Ce (Pr) SC epitaxial structures. LY—relative photoelectron light yield of SCFs under α-particle excitation in comparison with the LY of YAG:Ce SCF standard samples with a LY of 2.65 ph/keV. h-thickness of single crystalline film, f—velocity of SCF growth.

Sample Number	Samples	h (μm)	T (^o^C)	f (μm/min)	LY (%)
#1	YAP:Mn/YAP:Ce	10.5	983	0.12	14–37
#2	YAP:Mn/YAP:Ce	17	975	0.25	6
#3	(Lu_0.2_Y_0.8_)AP:Mn/YAP:Ce	52	980	0.7	6
#4	YAP:Mn/YAP:Pr	14	975	0.11	7.4
#5	GdAP:Ce/YAP:Ce	19	985	0.1	7
#6	GdAP:Ce/YAP:Ce	35	975	0.16	7
#7	GdAP:Ce/YAP:Pr	12	980	0.1	12

**Table 2 materials-15-08481-t002:** Properties of the investigated samples.

Material	Density of SCF (g/cm^3^)	Sample Number	SCF Thickness (µm)	10 keV Photon Absorption in SCF, (%)
YAP:Mn SCF/YAP:Ce SC	5.35	#1	10.5	21
		#2	17	32
LuYAP:Mn SCF/YAP:Ce SC	5.64	#3	25	59
YAP:Mn SCF/YAP:Pr SC	5.35	#4	14	27
GdAP:Ce SCF/YAP:Ce SC	6.09	#5	19	90
		#6	35	98
GdAP:Ce SCF/YAP:Pr SC	6.09	#7	12	77

**Table 3 materials-15-08481-t003:** The sets of peaks from sample #1, #2, #3, and #4 curves after irradiation by X-ray source or excitation by β-particles of ^90^Sr/^90^Y source.

**(#1) YAP:Mn SCF/YAP:Ce SC h = 10.5 μm**				**(#2) YAP:Mn SCF/YAP:Ce SC h = 17 μm**			
**^90^Sr/^90^Y 100Gy**		**X-rays 100Gy**		**^90^Sr/^90^Y 100Gy**		**X-rays 100Gy**	
**Peak** **(°C)**	**Energy (eV)**	**Peak** **(°C)**	**Energy (eV)**	**Peak** **(°C)**	**Energy (eV)**	**Peak** **(°C)**	**Energy (eV)**
-	-	54	1.6	-	-	62	0.7
81	0.7	75	1.3	74	0.8	82	0.5
145	1.4	143	1.1	143	1.4	137	1.4
162	2.4	167	1.3	155	1.2	156	1.6
189	1.7	193	1.2	190	1.3	196	0.6
213	1.4	-	-	224	1.2	-	-
259	2.4	249	3.0	257	2.9	241	0.6
318	1.3	315	1.25	310	1.2	299	1.0
**(#3) LuYAP:Mn SCF/YAP: Ce SC h = 25** **μ** **m**				**(#4) YAP:Mn SCF/YAP:Pr,Ce SC h = 14** **μ** **m**			
** ^90^ ** **Sr/^90^Y 100Gy**		**X-ray 100Gy**		** ^90^ ** **Sr/^90^Y 100Gy**		**X-ray 100Gy**	
**Peak** **(°C)**	**Energy (eV)**	**Peak** **(°C)**	**Energy (eV)**	**Peak** **(°C)**	**Energy (eV)**	**Peak** **(°C)**	**Energy (eV)**
-	-	84	1.7	-	-	87	0.6
80	1.7	90	1.4	68	1.05	145	0.5
139	1.6	140	1.6	82	1.2	150	1.1
152	2.3	183	0.9	142	1.4	192	1.5
182	2.8	238	1.6	174	2.2	242	1.7
235	1.5	-	-	237	1.4	-	-
251	2.5	274	2.4	252	2.8	259	2.9
304	0.8	293	1.6	305	1.03	306	1.1

**Table 4 materials-15-08481-t004:** The sets of peaks from sample #5, #6, and #7 curves after irradiation by X-rays or excitation by β-particles of ^90^Sr/^90^Y.

(#5) GdAP:Ce SCF/YAP:Ce SC h = 19 μm				(#6) GdAP:Ce SCF/YAP:Ce SC h = 35 μm				(#7) GdAP:Ce SCF/YAP:Pr SC h = 12 μm			
^90^Sr/^90^Y 100Gy		X-rays 100Gy		^90^sr/^90^y 100gy		X-rays 100gy		^90^Sr/^90^Y 100Gy		X-rays 100Gy	
Peak	Energy (eV)	Peak	Energy (eV)	Peak	Energy (eV)	Peak	Energy (eV)	Peak	Energy (eV)	Peak	Energy (eV)
182	1.8	182	1.7	82	1.6	79	0.4	79	1.5	80	1.3
145	1.4	145	1.4	147	0.4	151	1.4	141	1.2	145	1.2
241	1.5	-	-	244	1.5	-	-	235	1.5	-	-
256	2.0	254	1.6	263	2.0	254	1.6	255	2.0	258	1.5
301	1.9	300	2.4	302	1.9	306	1.9	292	1.8	306	2.4
329	1.8	328	1.9	331	2.5	331	2.4	327	1.45	327	1.4

## Data Availability

Not applicable.

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
