# Peer review of "Development of the Composite Thermoluminescent Detectors Based on the Single Crystalline Films and Crystals of Perovskite Compounds"

_materials, 2022, doi:10.3390/ma15238481_

Round 1
Reviewer 1 Report
Authors have fabricated composite thermoluminescent detectors using single crystalline films and single crystal perovskite. Single crystalline films and single crystal was fabricated by liquid phase epitaxy. This work looks interesting, but the authors failed to explain the motivation and importance of the research. Also, they didn't describe the synthesis/fabrication process as well as how the composite thermoluminescent works.
I would like to request a rewrite of the full manuscript in a proper way. I have several suggestions:
1. Please enrich the introduction part with proper mention of background and the motivation of your work.
2. You should properly describe the method of TL fabrication with schematics of the how the TL works.
3. Authors should use simple sentence structures instead complex structures.
4. Page 1 lines 12 and 13 "single crystalline films of Ce3+ and Mn3+ doped (Gd,Y)AlO3 ((Gd,Y)AP) perovskites and Ce3+ and Pr3+ doped YAlO3 (YAP) single crystal substrates" are they properly denoted?
5. In Figure 1, the authors should use a different background.
6. Authors should include compositional analysis results of the films or crystals using XPS or another proper method.
Author Response
This work looks interesting, but the authors failed to explain the motivation and importance of the research. Also, they didn't describe the synthesis/fabrication process as well as how the composite thermoluminescent works.
I would like to request a rewrite of the full manuscript in a proper way. I have several suggestions:
- Please enrich the introduction part with proper mention of background and the motivation of your work.
Thank you for your remark. We have tried to enrich the Introduction part of the paper. Namely, we have added new Figure 1 which describes operation of the composite TL detector. The motivation and importance of the research were extendedly described in the Introduction section.
- You should properly describe the method of TL fabrication with schematics of the how the TL works.
Basics of the thermoluminescence process has been added to chapter 3.2 ,,Thermoluminescence”.
- Authors should use simple sentence structures instead complex structures.
Thank you for your remark. Some sentences have been corrected.
- Page 1 lines 12 and 13 "single crystalline films of Ce3+ and Mn3+ doped (Gd,Y)AlO3((Gd,Y)AP)perovskites and Ce3+ and Pr3+ doped YAlO3 (YAP) single crystal substrates" are they properly denoted?
Corrected
- In Figure 1, the authors should use a different background.
Thank you for your comment regarding the picture., We will try to provide a more legible background in the next publications. This background aimed to show the transparency of the samples.
- Authors should include compositional analysis results of the films or crystals using XPS or another proper method.
Thank you for your remark, this is important information.
The concentration of dopants in substrate and SCF samples was measured using a SEM JEOL JSM-820 electron microscope equipped with a microanalyzer EDS with IXRF 500 and LN2 Eumex detectors. The Ce content in YAP:Ce and YAP:Pr crystal substrates was 0.25 % and 1 at. %, respectively. The Ce and Mn content in the SCF samples shows the proportional dependence on 1/Tg value, where Tg is temperature of film growth. Namely, the Ce3+ concentration in GdAP:Ce SCF was changed in the 0.065-0.01 at. % range depending on growth temperature Tg in the 975-985oC range (Table 1). The Mn content in YAP:Mn and (Y0.8Lu0.2AP) SCFs was 0.01-0.014 at. % and 0.016 %, respectively, depending on growth temperature Tg in the 975-985oC range (Table 1).
This paragraph is included in the revised version of manuscript.
Reviewer 2 Report
Referee report on manuscript “Development of the composite thermoluminescent detectors based on the single crystalline films and crystals of perovskite compounds”
This version does not look worthy and cannot be recommended for publication in this form and at least needs major revision.
1. Introduction. Relevance, novelty and relevant interest for a wide range of readers raises serious concerns.
2. Absolutely excessive self-citation is contrary to all ethical standards. Of the 13 cited papers, citations 1-10, 12 refer to the authors of this article, while no more than 25% would be reasonable.
3. After all, it is clear to everyone that these materials are interesting and promising and are the object of study by many research groups in different countries (Japan, Latvia, Kazakhstan, Estonia etc). See, some examples:
a) Nakanishi, K., et al. (2021). Performance evaluation of YAlO3 scintillator plates with different Ce concentrations. Applied Radiation and Isotopes, 168, 109483.
b) Piskunov, S.; et al. First Principles Calculations of Atomic and Electronic Structure of Ti3+Al- and Ti2+Al-Doped YAlO3. Materials 2021, 14, 5589. https://doi.org/10.3390/ma14195589
c) Karipbayev, Z.T.; et al Time-resolved luminescence of YAG:Ce and YAGG:Ce ceramics prepared by electron beam assisted synthesis. Nucl. Instrum. Methods Phys. Res. B. Beam Interact. Mater. Atoms 2020, 479, 222–228.
Therefore, the authors are encouraged to revise the introduction and give a broader view of the study of these materials.
4. In the conclusions, it is necessary to clearly formulate what new data about the studied materials were obtained in this work?
In general, the manuscript is interesting and can be considered for publication after constructive reflection on the above comments.
Author Response
- 1.Relevance, novelty and relevant interest for a wide range of readers raises serious concerns.
Thank you, we have completely rewritten the Introduction with the aim of more clear representation of the new approaches, ideas and results in comparison with previous works of our group in the frame of development of composite scintillators and TL detectors using LPE growth method. This work is very important step in the development of multifunctional detectors of ionization radiation suitable for environmental monitoring, nuclear physics and medicine(oncology), based on the different oxide compounds. In this regards, RE and transition metal doped perovskites belong to suitable and prospective compositive materials in both crystal and film forms.
- Absolutely excessive self-citation is contrary to all ethical standards. Of the 13 cited papers, citations 1-10, 12 refer to the authors of this article, while no more than 25% would be reasonable. After all, it is clear to everyone that these materials are interesting and promising and are the object of study by many research groups in different countries (Japan, Latvia, Kazakhstan, Estonia etc). See, some examples:
- a)Nakanishi, K., et al. (2021). Performance evaluation of YAlO3scintillator plates with different Ce concentrations. Applied Radiation and Isotopes, 168, 109483.
- b)Piskunov, S.; et al. First Principles Calculations of Atomic and Electronic Structure of Ti3+Al- and Ti2+Al-Doped YAlO3. Materials 2021, 14, 5589. https://doi.org/10.3390/ma14195589
- c)Karipbayev, Z.T.; et al Time-resolved luminescence of YAG:Ce and YAGG:Ce ceramics prepared by electron beam assisted synthesis. Instrum. Methods Phys. Res. B. Beam Interact. Mater. Atoms2020, 479, 222–228.
Therefore, the authors are encouraged to revise the introduction and give a broader view of the study of these materials.
We are agree with this comment. The list of references has been supplemented with appropriate papers.
- In the conclusions, it is necessary to clearly formulate what new data about the studied materials were obtained in this work?
Thank you. We have completely rewritten the Conclusion in the frame of such Referee’s recommendations.
Reviewer 3 Report
Dear Authors
The paper presents interesting results and it is well written. I think it is ready for publication after minor amendments. There are few textual comments from my side that should help improving the text and are easy to implement
My congratulations!
General
Abstract: please, do not use abbreviations in the abstract as it complicates reading. Just move it to the main text at the first place they are introduced.
Captions: Please, do not use abbreviations in the figure/table captions.
Reference: I think there are should be much more papers to cite on this vibrating subject, please, give sufficient credits to your colleagues. The current list is too short and quite biased towards to publications of the scientist from the paper author-list.
Figure referencing in the text: sometimes you use Figure X, sometime Fig.X, please choose one style.
Detailed
L32: Citation [1-5;6-7] should be [1-7]. And it is really too compressed. Since it is introduction, it is quite important to a reader to understand what is special in each cited publication and if one should also read that one. Please, mention each paper separately and give few words of description.
L52: “after alpha/beta particle excitation” : it is very unclear here why these particles are excited. From a later description it is getting clear that the materials are irradiated by alpha/beta particles… Please, rephrase the text.
L52: For characterize of -> to characterize
L52: rate of alpha/beta particle separation: not very clear what is separated, particles or the rate. Please, rewrite this part.
L53: delta T: not clear what the T is and in what units it is measured.
L57: positions (up to 80 deg): please, explain better the setup. I assume a position to be an observable measured in meters…
L61: x=2-1.5 : what is x and what are units?
L77: to noted -> to note (or to be noted)
L80: also possess -> possesses
L81: try to connected -> try to connect
L90: 0.25-0.5 at. % : strange units at.%
L101: n.m. not measured : not sure what is not measured and why it is mentioned in the caption
Table 1: please, introduce h and f
L107: As can see -> as can be seen (or as one can see)
L107: much better -> a much better
L112: Mn doped -> Mn-doped
L136: “All of the …” : not clear what do you mean… Perhaps “Some of the…” or “Few most important for the current research…”
L137: “was calculated by the formula 1” -> “was calculated using” and move the sentence starting with “Where mu..” after the equation.
L155: present -> is
L162: intensive -> intensity
L165 and 167 (and few others): 254 and 275 nm -> 245 nm and 275 nm
Fig 3 caption: picture (b) is not explained
L176: TL is already introduced
L208: x-ray -> X-ray
L215 Table 2 : please, use non-breakable space between “Table” and “2” to avoid separation on two different lines (same on L236)
L235: dived -> divided?
L244 the differences well -> the differences
L267: aimed -> aims
L273: CL and TL are already introduced
Author Response
Abstract: please, do not use abbreviations in the abstract as it complicates reading. Just move it to the main text at the first place they are introduced.
Captions: Please, do not use abbreviations in the figure/table captions.
Thank you for your recommendation. The abstract and figure/table captions have been corrected
Reference: I think there are should be much more papers to cite on this vibrating subject, please, give sufficient credits to your colleagues. The current list is too short and quite biased towards to publications of the scientist from the paper author-list.
The additional literature references have been added in the revised version of the manuscript .
Figure referencing in the text: sometimes you use Figure X, sometime Fig.X, please choose one style.
The style was unified in the revised version of manuscript. In parentheses (Fig. X), while in the text Figure X
L32: Citation [1-5;6-7] should be [1-7]. And it is really too compressed. Since it is introduction, it is quite important to a reader to understand what is special in each cited publication and if one should also read that one. Please, mention each paper separately and give few words of description.
Corrected
L52: “after alpha/beta particle excitation” : it is very unclear here why these particles are excited. From a later description it is getting clear that the materials are irradiated by alpha/beta particles… Please, rephrase the text.
Corrected
L52: For characterize of -> to characterize
Corrected
L52: rate of alpha/beta particle separation: not very clear what is separated, particles or the rate. Please, rewrite this part.
Corrected
L53: delta T: not clear what the T is and in what units it is measured.
Corrected
L57: positions (up to 80 deg): please, explain better the setup. I assume a position to be an observable measured in meters…
Corrected
L61: x=2-1.5 : what is x and what are units?
X is a part of formula: Lu3−xGdxAl5O12:Ce SCFs /YAG:Ce S.C.
L77: to noted -> to note (or to be noted)
Corrected
L80: also possess -> possesses
Corrected
L81: try to connected -> try to connect
Corrected
L90: 0.25-0.5 at. % : strange units at.%
Atomic percentage is a typical unit indication of dopant concentration. Such a unit can be directly obtained from the microanalysis of materials, performed using respective detectors at SEM.
L101: n.m. not measured : not sure what is not measured and why it is mentioned in the caption
Corrected
Table 1: please, introduce h and f
Corrected
L107: As can see -> as can be seen (or as one can see)
Corrected
L107: much better -> a much better
Corrected
L112: Mn doped -> Mn-doped
Corrected
L136: “All of the …” : not clear what do you mean… Perhaps “Some of the…” or “Few most important for the current research…”
Corrected
L137: “was calculated by the formula 1” -> “was calculated using” and move the sentence starting with “Where mu..” after the equation.
Corrected
L155: present -> is
Corrected
L162: intensive -> intensity
Corrected
L165 and 167 (and few others): 254 and 275 nm -> 245 nm and 275 nm
Corrected
Fig 3 caption: picture (b) is not explained
Corrected
L176: TL is already introduced
Corrected
L208: x-ray -> X-ray
Corrected
L215 Table 2 : please, use non-breakable space between “Table” and “2” to avoid separation on two different lines (same on L236)
Corrected
L235: dived -> divided?
Corrected
L244 the differences well -> the differences
Corrected
L267: aimed -> aims
Corrected
L273: CL and TL are already introduced
Corrected.
Round 2
Reviewer 1 Report
Thanks to the authors for their effort to improve the manuscript. After revision, the manuscript looks better, and I recommend accepting this current form.
Reviewer 2 Report
After revision, this manuscript looks OK and can be recommended for publication